# Diatom lipids open window to past ocean temperatures in the polar regions
Simon T. Belt [1,2] ✉, Lukas Smik[3], Denizcan Köseoğlu[4], Claire S. Allen [5], Katrine Husum [6] &
Jochen Knies [7,8]

Sea surface temperature is a key indicator of climate change on Earth and is central to all related modelling endeavours. However, sea surface temperature is notoriously difficult to reconstruct accurately in the geological record, especially for the low temperatures of the polar regions, which occupy one-third of the world's oceans. Here we show that a sea surface temperature proxy based on two isomeric diatom lipid biomarkers can be applied to marine sediment archives to reconstruct temperatures in the range −1 to 14 °C for the Arctic and Antarctic using a single calibration. For both regions, our datasets span timeframes from recent decades to the Younger Dryas/Holocene, and we also showcase a 750 kyr record from the Fram Strait, the major gateway between the North Atlantic and the Arctic Ocean. We anticipate that this lipid biomarker-based proxy may become a standard component of the palaeoclimate toolkit, especially for the polar regions.

The reconstruction of ocean temperatures is pivotal for understanding Earth's climate history. Sea surface temperature (SST) is a key variable in the modelling of both past and future climate states[1] and its determination in the geological record is commonly achieved through application of biotic and geochemical proxies archived in marine sediment cores. Representing around 30% of the world's oceans[2], the polar regions are central to Earth's climate, more generally. Rising ocean temperatures in high latitude regions are leading to reductions in sea ice, accelerated glacial melting, sea-level rise, reduced solubility of greenhouse gases (e.g., $CO_2$), increased evaporation, as well as perturbations to marine ecosystems[3–8]. The current trend of rising global temperatures worldwide is amplified in the polar regions[9], yet it is here that many SST proxies are least reliable. For the $U_{37}^K$ palaeothermometer, which is based on the temperature-dependent distribution of $C_{37}$ alkenone lipids made by haptophyte algae[10], and is the most widely used SST proxy worldwide, the index is insensitive to temperatures below ca. 8 °C, according to an early global core top calibration[11]. On the other hand, some improvements in extending the linear range have been made in recent years, including the use of $C_{38}$ alkenone-based indices[12,13]. Further, the $TEX_{86}$ index, derived from distributions of isoprenoid glycerol dialkyl glycerol tetraethers (isoGDGTs) has a temperature response that flattens at ca. 12 °C[14] and most likely reflects sub-surface temperatures rather than SSTs[15]. However, some recent developments and the introduction of related proxies based on hydroxylated isoGDGTs appear to be improving the reliability of isoGDGT-

based temperature proxies in the polar regions[16–20]. For both alkenone- and isoGDGT-based temperature proxies, absolute and relative temperature reconstructions can be strongly dependent on the calibration used (see later), and needs be taken into account, not least when making comparisons between individual studies. More generally, some SST proxy source biota in the polar regions are sometimes not sufficiently widespread, abundant or preserved for their fossilised remains (including lipid biomarkers) to be easily detected in marine sediments[21,22], and further complications exist for biotic and geochemical proxies regarding depth habitats, seasonality, advection and other non-thermal environmental factors such as salinity and sea ice[19,23–28]. Indeed, an Arctic sea-ice (i.e., sympagic) alkenone source has been identified recently[29], whose distribution of alkenones is independent of temperature, at least in culture, and produces relatively large quantities of the $C_{37:4}$ alkenone, which frequently results in anomalies in $U_{37}^K$-based SST reconstructions[30,31]. The same sympagic alkenone source also yields enhanced production of the $C_{37:3}$ alkenone (relative to $C_{37:2}$), with consequential lowering of SSTs, irrespective of which alkenone unsaturation index calibration is used. In fact, this contribution of sympagic alkenones is likely a major cause of the inaccuracy of $U_{37}^K$-based SSTs below ca. 8 °C in the Arctic, at least[30,31].

In a recent study, we showed that the relative amounts of two tri-unsaturated and isomeric highly branched isoprenoid (HBI) biomarkers (i.e. HBIs III and IV; Fig. 1) derived from common diatoms, such as *Rhizosolenia*, were well correlated with in situ temperature in water samples

[1]Biogeochemistry Research Centre, University of Plymouth, Plymouth, UK. [2]SBelt25 Consulting, Ivybridge, Devon, UK. [3]Centre for Resilience in Environment, Water and Waste, College of Life and Environmental Sciences, University of Exeter, Exeter, UK. [4]AltraFlora Natural Extracts Inc. Merkezefendi, Denizli, Turkey. [5]British Antarctic Survey, High Cross, Madingley Road, Cambridge, UK. [6]Norwegian Polar Institute, Fram Centre, Tromsø, Norway. [7]Geological Survey of Norway, Trondheim, Norway. [8]iC3: Centre for ice, Cryosphere, Carbon and Climate, Department of Geosciences, UiT The Arctic University of Norway, Tromsø, Norway.
✉e-mail: sbelt@plymouth.ac.uk; s.belt25@gmail.com

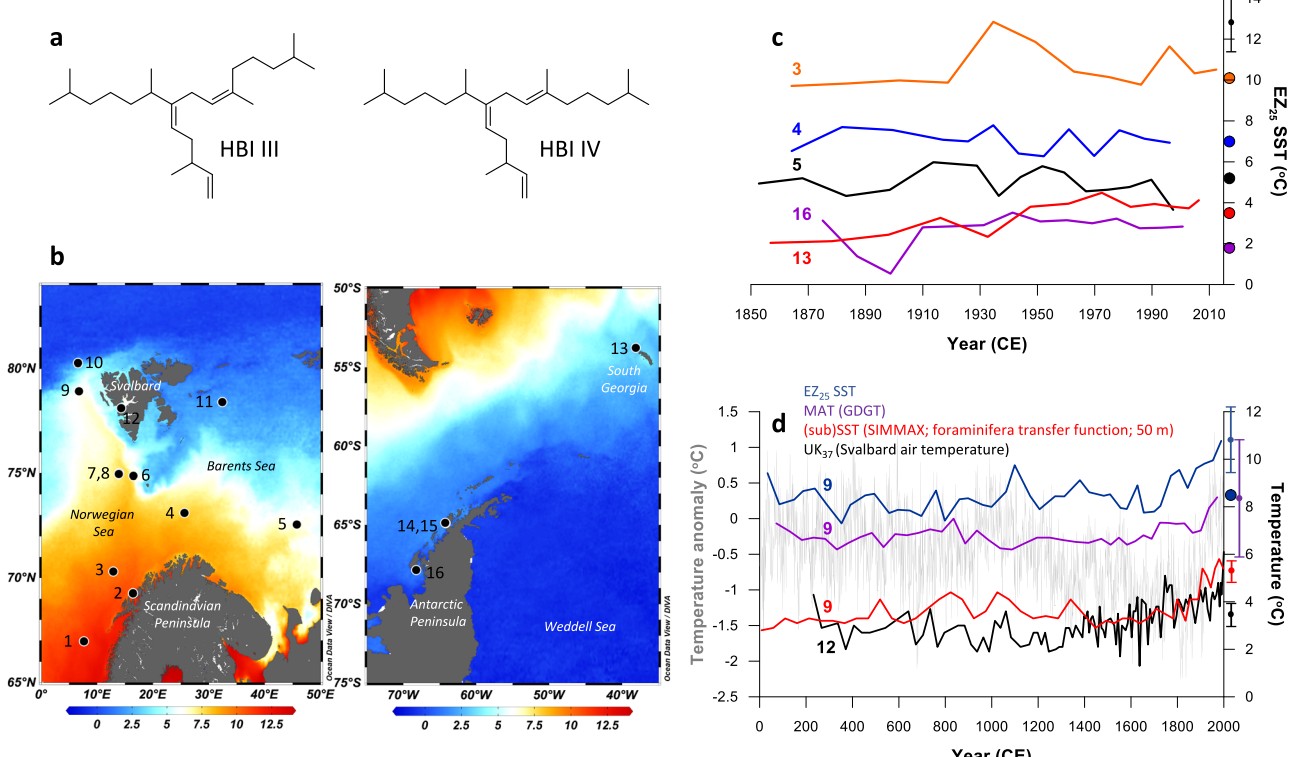

**Fig. 1 | Structures, locations of marine sediment cores and EZ$_{25}$ sea surface temperature (SST) reconstructions over recent decades/centuries. a** Structures of highly branched isoprenoid (HBI) biomarkers used for the EZ$_{25}$ SST proxy; **b** Summary maps showing the locations of the marine sediment cores described in the current study. Core names indicated by core numbers are as follows: 1. MD95-2011 (66.97°N, 7.63°E); 2. JM99-1200 (69.27°N, 16.42°E); 3. R248 MC010 (70.31°N, 12.88°E); 4. BASICC 1 (73.10°N, 25.63°E); 5. BASICC 43 (72.54°N, 45.74°E); 6. JM09-KA11-GC (74.87°N, 16.48°E); 7. M23258 (75.00°N, 13.97°E); 8. SV04 (74.96°N, 13.90°E); 9. MSM5/5-712-1 (78.92°N, 6.77°E); 10. 910 A (80.26°N, 6.59°E); 11. NP05-11-70GC (78.4°N, 32.42°E); 12. Kong-B (78.11°N, 14.30°E); 13. BC 660 (53.78°S, 38.13°W); 14. ODP 1098 (64.87°S; 64.23°W); 15. JPC-10 (64.88°S, 64.20°W); 16. BC 523 (67.86°S, 68.20°W); **c** EZ$_{25}$-based SST records spanning the past ca. 150 yr from locations in the Arctic (cores 3 (R248 MC010; orange line), 4 (BASICC 1; dark blue line) and 5 (BASICC 43; black line)) and the Antarctic/ Southern Ocean (cores 13 (BC 660; red line) and 16 (BC 523; purple line)). Coloured dots on the SST scale refer to modern Spring/Summer SSTs at the core locations (see main text for details). The standard error for the EZ$_{25}$ SST proxy (± 2.8 °C) based on a previous water column calibration[32] is shown as a black vertical error bar symbol; **d** summary compilation of temperature records for Western Svalbard over the past ca. 2,000 yr: EZ$_{25}$-based SST (core 9; MSM5/5-712-1; dark blue line); GDGT-based mean air temperature (core 9; MSM5/5-712-1; purple line)[38]; SIMMAX (foraminifera transfer function) sub-surface (50 m) temperature (core 9; MSM5/5-712-1; red line)[35]; alkenone-based mean June–August air temperature (core 12; Kong-B; black line)[39]; pan-Arctic 2k temperature anomaly (light grey line[40]). The dark blue dot refers to the maximum (summer) SST at the core location (see main text for details). The standard errors for the different proxies are shown as vertical error bar symbols using the same colours as the corresponding data plots. Maps were produced using Ocean Data View (source: http://odv.awi.de/).

taken from the Arctic, the Antarctic and the western English Channel, within a temperature range of ca. –1 to 18 °C[32]. Thus, the EZ$_{25}$ index (Eq. 1) was found to be linearly correlated with in situ SST (Eq. 2) with a standard error of ± 2.8 °C. In order to investigate whether the EZ$_{25}$ index could also be used as a proxy for SST in the palaeo record, in the current study we applied this linear correlation (Eq. 2) to concentrations of the same biomarkers in marine archives from the Arctic and Antarctic. Our findings demonstrate that the EZ$_{25}$ index can indeed provide realistic estimates of past SSTs for both polar regions using a single calibration, and over decadal to centennial timescales spanning the last ca. 750 kyr, at least.

$$EZ_{25} = \frac{[HBI\ IV]}{([HBI\ III] + [HBI\ IV])} \quad (1)$$

$$SST = \frac{(EZ_{25} - 0.169)}{0.032} \quad (2)$$

## Results and Discussion
### Evaluating EZ$_{25}$ SST records over recent decades and centuries
In order to test the utility of EZ$_{25}$ as a SST proxy for the polar regions, we first measured it in a collection of short cores from the Arctic and Antarctic for

which the corresponding past SSTs over recent decades/centuries are constrained by satellite and other direct observations, together with other geologic SST proxy datasets (Fig. 1b–d). For example, for the Arctic over recent timeframes (the past ca. 150 yr) reconstructed EZ$_{25}$-derived SSTs from a short sediment core (R248 MC010; core 3; Fig. 1b, c) located within the main axis of the North Atlantic Current (NAC) in the Norwegian Sea shows a mean value (10.6 °C) that aligns closely with mean spring/summer SSTs for the region (10.1 °C from 2003–2022[33]). For locations further east, the diminished influence of the NAC and increased contribution from Arctic surface waters is reflected in a reduction in mean reconstructed SSTs from short cores BASICC 1 (7.1 °C) and BASICC 43 (5.0 °C) (cores 4 and 5; Fig. 1b, c), again in close agreement with those of mean spring/summer months (7.0 °C and 5.2 °C, respectively, from 2003–2022[33]), the main period of phytoplankton growth in the region[34].

EZ$_{25}$-based SSTs from a further short core from the western Svalbard margin (MSM5/5–712–1; core 9; Fig. 1b, d) are, in contrast, more variable, exhibiting trends that closely follow (sub)surface (ca. 50–150 m) temperatures derived from planktic foraminifera and Mg/Ca ratios[35], albeit with a warm temperature offset for the EZ$_{25}$-based SSTs, consistent with warmer summer surface waters (Fig. 1d). The core-top SST from MSM5/5–712–1 (10.8 °C) is slightly higher than contemporary summer SSTs at the core site (e.g., ca. 8.5 °C[35]), yet the difference (2.3 °C) is, nonetheless, within the standard error (SE) of the EZ$_{25}$ proxy (2.8 °C[32]), so the temporal trends are

**Fig. 2 | Compilation of temperature records for Western Svalbard over the past ca. 2,000 yr.** **a** Alkenone-based SST records from core 9 (MSM5/5-712-1). The brown line corresponds to the $U^K_{37}$-based SSTs[38]. The orange line corresponds to a revised $U^{K\prime}_{37}$-based SST record, which excludes the $C_{37:4}$ alkenone (likely from sea ice[29]) in the calculation and yields more realistic SSTs at the core location (The maximum modern summer SST is indicated with a black dot); **b** %$C_{37:4}$ (alkenone) content in core 9; MSM5/5-712-1 (ref. 38); **c** $EZ_{25}$-based SST (core 9; MSM5/5-712-1; dark blue line); GDGT-based mean air temperature (core 9; MSM5/5-712-1; purple line[38]); SIMMAX (foraminifera transfer function) sub-surface (50 m) temperature (core 9; MSM5/5-712-1; red line[35]); alkenone-based mean June-August air temperature (core 12; Kong-B; black line[39]); pan-Arctic 2k temperature anomaly (light grey line[40]). In each case, the proxy standard error is indicated with a vertical error bar symbol using the same colour as the corresponding data plot.

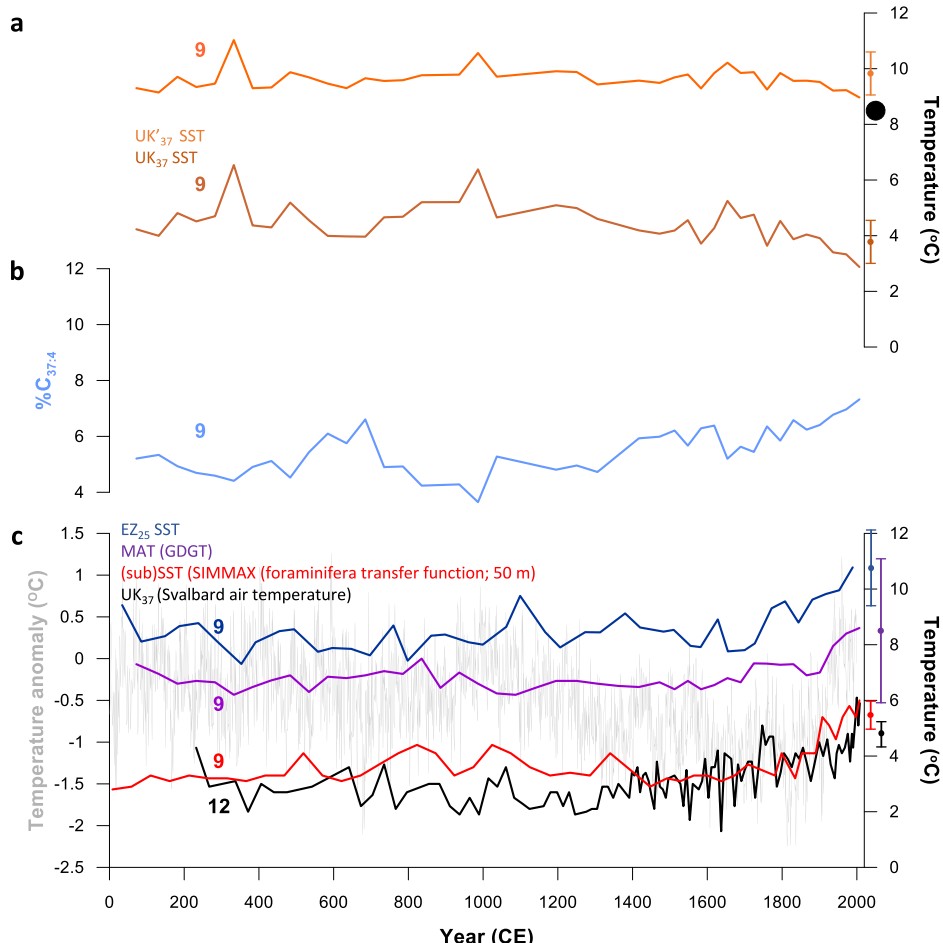

considered reliable. Of particular note is the sharp increase in $EZ_{25}$-based SST (and subSSTs from forams and Mg/Ca[35]) during recent centuries, attributed previously to enhanced northward heat flux derived from the NAC[35], and consistent with a well-established decline in seasonal sea ice during the same interval from both observational and proxy records[36,37], two independent air temperature records for Svalbard[38,39] and the pan-Arctic temperature trend over the last 2 kyr[40], more generally (Fig. 1d). On the other hand, an apparent cooling trend in SST has been reported based on alkenone palaeothermometry (MSM5/5–712–1[38]). Notably, the published alkenone-derived SST profile (Fig. 2a) closely mirrors the %$C_{37:4}$ alkenone content throughout the record (Fig. 2b) so the apparent decline in SST towards the core-top is thus likely an artefact driven by contributions from sympagic alkenone sources[29]. Indeed, a revised alkenone SST reconstruction omitting the $C_{37:4}$ contribution shows a negligible decline and more realistic SSTs overall (Fig. 2a).

We also investigated the potential of $EZ_{25}$ as a low temperature SST proxy in two short cores from the Southern Ocean; one from the West Antarctic Peninsula (WAP) and a further core from South Georgia, each representing the last century or so (Fig. 1b). $EZ_{25}$-based SSTs from box core 523 (BC 523; core 16; Figs. 1b, c, 3a) from Ryder Bay (WAP) yield a mean value of 2.7 °C ± 0.8 °C for the past ca. 120 yr. SSTs at the core site peak during January (mean 1.8 °C[33]) although in situ SSTs of 2.5 °C–3.5 °C have been recorded between December and February at the Rothera Annual Time Series station, close (ca. 30 km) to where BC 523 was collected[41]. A single negative spike in SST is evident ca. 1900 CE and is coeval with an increase in the ratio $IPSO_{25}$/HBI III, a qualitative measure of sea ice extent based on the Antarctic sea ice proxy $IPSO_{25}$[42] and HBI III from pelagic diatoms[43,44] (Fig. 3a). As such, an acute drop in SST likely indicates a period of protracted sea ice cover and reduced open water conditions in the embayment at this time, an interpretation corroborated by observational

records at the core region, which show an inverse correlation between spring/summer SSTs and sea ice extent[45].

For South Georgia, which currently experiences year-round open waters free of sea ice, our reconstructed SSTs for box core 660 (BC 660; core 13; Fig. 1b) reveal a steady increase from ca. 2.0 °C–4.0 °C beginning in the mid-19th century (Figs. 1c, 3b). These SSTs and the overall warming trend agree well with instrumental records, which exhibit an increase in SST around South Georgia of ca. 1.5 °C since the 1920s (i.e., from 2.0 °C to 3.5 °C[46]). In fact, mean (instrumental) temperature increases (0.13 °C and 0.16 °C/decade for Oct–Mar and Annual, respectively; 0–50 m water depth[46]) are in close alignment with that derived from $EZ_{25}$ (0.14 °C/decade) since the mid-19th century (Fig. 3b).

In summary, on the basis of the datasets presented here, $EZ_{25}$ provides realistic SSTs from recent Arctic and Antarctic marine sedimentary archives, at least when compared with independent measures, including direct observation.

## SST records over millennial to orbital timescales

As a next step, we measured $EZ_{25}$ in a further suite of cores representing timeframes that extend beyond the observational record in order to test whether the corresponding SSTs were as expected based on previous environmental proxy data, or could provide SST data where previous estimates were unavailable. In the Arctic, the sensitivity of the Norwegian/Barents Seas to longer-term environmental change can be seen through temporal records of $EZ_{25}$-based SSTs in downcore records from sites at the confluence of Atlantic and Arctic waters (i.e., the Barents Sea Arctic Front (BSAF) or Polar Front (PF)), and complementary locations to the west and east. Sediment core JM99–1200 from the Andfjorden in northern Norway (core 2; Fig. 1b) contains a marine climate record from ca. 14.0–7.0 kyr BP; i.e. the Bølling–Allerød warm interval (B–A; ca. 14.0–12.9 cal. kyr BP), the

**Fig. 3 | EZ25-based SSTs for the Southern Ocean representing the last century or so. a** Core 16 (BC 523; Ryder Bay; WAP). The ratio IPSO25/HBI III (red line) is used as a qualitative measure of directional changes in seasonal sea ice cover[44]. The change in IPSO25/HBI III is mirrored by changes in EZ25 SST (dark blue line) consistent with observational records from Ryder Bay[45]. **b** Core 13 (BC 660; South Georgia). The mean decadal increase in SST (0.14 °C/decade) aligns well with those measured in observational records close to the core site (0.13 °C and 0.16 °C/decade for Oct–Mar and Annual, respectively; 0–50 m water depth[46]). The standard error of the EZ25 SST proxy (2.8 °C) from an earlier water column calibration[32] is indicated by a vertical error bar symbol.

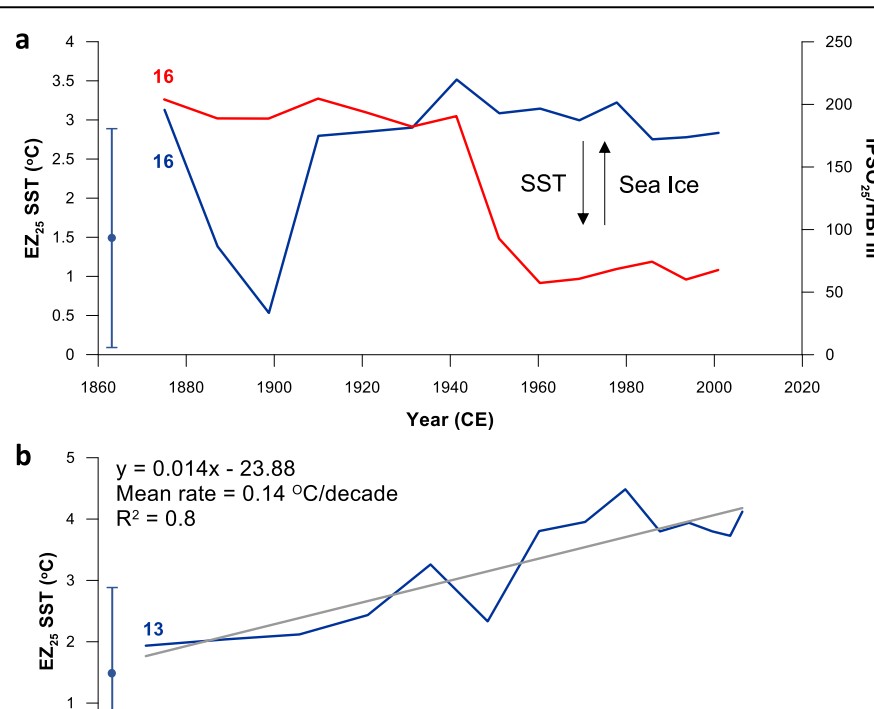

Younger Dryas cold stadial (YD; ca. 12.9–11.9 cal kyr BP) and the early Holocene (ca. 11.9–7.0 cal kyr BP). Previous proxy records show that the core site was covered in seasonal sea ice cover during the YD only, with year-round ice-free conditions prevailing before and after[47]. Consistent with these two contrasting surface ocean states, EZ25-based SSTs decline abruptly by over 5 °C from (mean values) 13.2 °C during the B–A, to 7.9 °C for the YD (Figs. 4a, 5b), coincident with the occurrence of seasonal sea ice, as shown by the presence of the Arctic sea ice biomarker IP25 (Fig. 5a)[47], before recovering to 12.3 °C at the beginning of the early Holocene (11.9–10.0 cal kyr BP), when the surface ocean returned rapidly to ice-free conditions (absent IP25). These findings are supported by other (partial) data from JM99–1200 based on planktic foraminifera assemblages[48] and alkenones ($U^{K}_{37}$)[49] (Fig. 4a) together with early Holocene alkenone-based SSTs from nearby MD95–2011[50] (core 1; Figs. 1b, 5). The ca. 5 °C–6 °C drop in EZ25-based SSTs from JM99-1200 during the YD mimics the change in SST between sea-ice covered and ice-free locations in the Nordic and Barents Seas in modern times, especially during spring–autumn months (i.e. during active diatom growth), while the mean YD SST (7.9 °C) aligns well with the warm end of seasonally ice-covered locations in the modern Barents Sea (maximum ca. 8.7 °C[2]). Unlike our continuous EZ25 SST record in JM99–1200, alkenones were largely absent in JM99–1200 during the YD, and alkenone-based SSTs in MD95–2011 throughout the YD are unreliable due to the high content of the C37:4 alkenone[50] (Fig. 5b), likely derived from sea ice sources (c.f. MSM5/5–712–1; see above).

A near-identical pattern in EZ25-based SSTs in JM99–1200 is evident in core JM09–KA11–GC (western Barents Sea; core 6; Figs. 1b, 5c), a site also characterised by extensive sea ice during the YD[51]. Mean SSTs for the B–A, YD and early Holocene (11.0 °C, 8.7 °C and 11.0 °C, respectively (Fig. 5c)), are slightly lower (1 °C–2 °C) compared to JM99–1200 during the B–A and early Holocene reflecting its more northerly location (Fig. 1b). These SST changes observed in JM99–1200 and JM09–KA11–GC highlight the abrupt temporal shifts in the position of the BSAF, most notably around the time of the YD[52]. Following early Holocene peak warmth, SSTs decline thereafter in

JM09–KA11–GC, likely driven by reduced solar insolation, with late Holocene values (ca. 5.5 °C) close to modern spring (Apr–June) SSTs at the core site (6.2 °C[33]; Fig. 5c).

EZ25-based SSTs exhibit a different pattern in core M23258 (core 7; Figs. 1b, 5d), located west of JM09–KA11–GC, with only a short-term drop during the YD (ca. 12.3 cal. kyr BP) since the site only experienced minimal sea ice conditions at that time due to its position somewhat to the west of the BSAF[52]. Further, our reconstructed SSTs during the Holocene align closely with those derived from alkenones from the same core[53] and from core SV04[54] (core 8; Fig. 1b; Fig. 5d), reflecting their locations along the main trajectory of the NAC (c.f. R248 MC010; see above). The YD section of M23258 contains elevated percentages of the C37:4 alkenone (from sea ice; Fig. 5d) so the corresponding alkenone-based SSTs are likely underestimated (c.f. MSM5/5–712–1 and MD95–2011; see above).

The occurrence of a relatively warm and stable SST profile in M23258 has a cold counterpart in the eastern part of the region, with SSTs based on EZ25 from core NP05–11–70GC (northern Barents Sea; core 11; Fig. 1b) persistently low throughout the Holocene (ca. 5.5 °C ± 1.0 °C; Fig. 5e) due to the continuous occurrence of seasonal sea ice[55], a prevalence of Arctic waters and no warming influence from the NAC. The late Holocene SST (5.3 °C) is close to contemporary late summer SSTs for the region (3 °C–4 °C[33]), reflecting delayed pelagic diatom growth stimulated by late-season sea ice retreat.

For the Antarctic, EZ25-based SSTs for the Holocene (0.8 °C ± 1.1 °C) from ODP site 1098 (Palmer Deep; WAP; core 14; Fig. 1b) sit well within the seasonal range at the core site (i.e., −1 °C to 2.5 °C[56]) and also with previously reported TEX86- and TEX$^{L}_{86}$-derived SSTs from the same core[56, 57] and a nearby site (JPC–10; core 15; Fig. 4b)[57]. Despite the low amplitude variability, an early Holocene increase in EZ25-based SSTs is evident (ca. 11 cal. kyr BP), and the mean values for the early (0.5 °C ± 1.4 °C), middle (-0.1 °C ± 0.6 °C) and late Holocene (-0.3 °C ± 0.6 °C) are consistent with a previously reported SST temperature trend[56,57]. We note, however, that the warmer early Holocene SSTs derived from EZ25 are not as high as those obtained using the TEX86 proxy. The reason for this is not clear at present

**Fig. 4 | EZ₂₅ SST reconstructions covering the Younger Dryas and Holocene in the Arctic and Antarctic. a** SST reconstructions for core 2 (JM99-1200) located in the Andfjorden (northern Norway; Fig. 1b). The dark blue line corresponds to the continuous EZ₂₅-based SST record, while the orange and red lines represent partial SST records derived from alkenones[49] and a foraminiferal transfer function[48], respectively. The dramatic decline in EZ₂₅-based SSTs during the Younger Dryas cold stadial (ca. 12.9–11.9 cal kyr BP) is bracketed by substantially warmer SSTs during the Bølling–Allerød warm interval and the early Holocene Thermal Maximum; **b** SST records from Palmer Deep (West Antarctic Peninsula; Fig. 1b): EZ₂₅-based SSTs from core 15 (ODP 1098; dark blue line), TEX₈₆ SST (0–150 m) record from core 15 (ODP 1098; grey line[56];) and TEX₈₆ᴸ SST record from core 14 (JPC-10; black line[57];). The three-division classification of the Holocene is taken from Etourneau et al.[57]. In each case, the proxy standard error is indicated with a vertical error bar symbol using the same colour as the corresponding data plot.

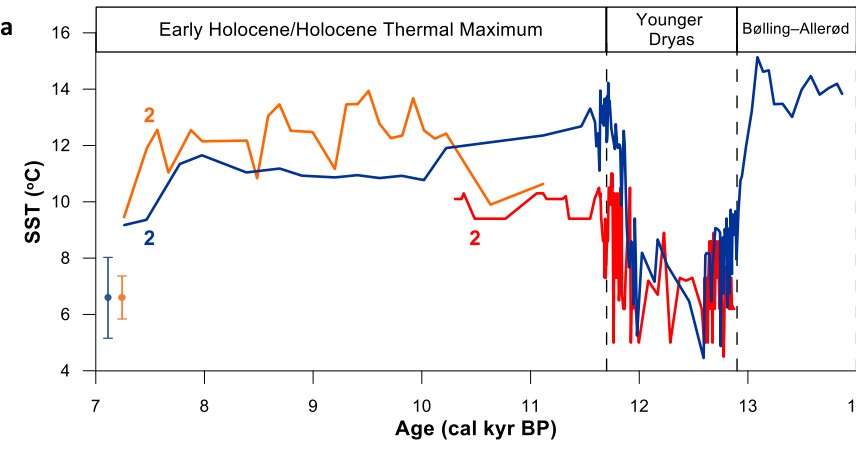

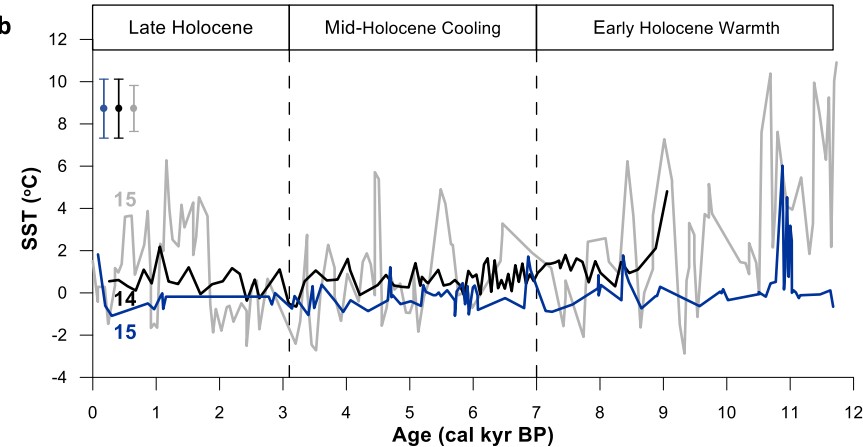

but could be due to inaccuracies associated with earlier TEX₈₆ calibrations[14,56,57] or differences in seasonal signatures between the two proxies.

Finally, to test the reliability of EZ₂₅-based SSTs over even longer timescales, we measured it in marine sediments covering the past 750 kyr from Ocean Drilling Programme (ODP) Hole 910 A, situated on the Yermak Plateau (NW Svalbard) at the Arctic-Atlantic gateway (AAG; core 10; Fig. 1b). The occurrence of the two constituent HBI biomarkers (viz. HBIs III and IV) and a mean SST spanning the last ca. 750 kyr of ca. 3.3 °C (Fig. 6b) supports the recent conclusion that, on sub-orbital to millennial timescales, the AAG experienced seasonal (i.e. late autumn-spring) sea ice conditions and open waters during late spring/summer months[58], with EZ₂₅ SST estimates resembling contemporary summer (July–September) values (3.0 °C[33]). Although there is no clear alternating trend in SST between glacial and interglacial cycles, with even some relatively warm and cold SSTs during certain glacials (e.g. MIS 4, 14) and interglacials (e.g. MIS 11), respectively, some of the lowest SSTs are observed during Marine Isotope Stage (MIS) 16 (mean SST = 2.9 °C ± 0.2 °C), consistent with other bioproductivity and δ¹⁸O (planktonic) proxy data[58] (Fig. 6e–g). Given the low overall variability in SST (mean ± SD = 3.3 °C ± 0.3 °C), we believe that at the sampling resolution investigated (ca. 1000–6500 yr[58]), SSTs at the core site were largely insensitive to glacial-interglacial change. Consistent with this, we also observe only small variations in the sea ice biomarker IP₂₅ throughout the record (Fig. 6d) and estimates of spring sea ice concentration (SpSIC (%); Fig. 6c) are also extremely uniform and close to the modern mean annual value (35%; ref. 59).

Collectively, the outcomes from these longer-term SST reconstructions suggest that the EZ₂₅-SST relationship, first identified in near-surface water column samples[22] and also in various short core records described herein, is preserved over extended geological intervals.

## Outlook

Based on the data presented here, we propose that the EZ₂₅ SST proxy has the potential to provide realistic SSTs in the temperature range ca. −1 °C to +18 °C from late Quaternary sediment sequences spanning the last ca. 750 kyr, at least, thus providing a possible solution to a long-standing limitation of some existing SST proxies, which are frequently not reliable for the cold ocean temperatures of the polar regions or are barren in marine sediment archives.

The EZ₂₅ proxy has potentially several other advantages over other biomarker-based ocean temperature proxies. Most notably, all of the Arctic and Antarctic temperature reconstructions described herein have been achieved using a single calibration, the calibration itself is based on surface water column samples, thus reflecting SSTs rather than sub-surface temperatures, and the residuals of reconstructed SSTs versus measured temperature (in situ or satellite-derived) are relatively small at low SSTs[32]. Further, the EZ₂₅ proxy is comprised of two co-produced biomarkers with almost identical structures, which likely mitigates against variability in overall production and differential biomarker degradation in the water column and sediments, respectively. Although there are relatively few known sources of the two constituent HBI biomarkers, one such genus, *Rhizosolenia*, is nonetheless ubiquitous in the world's ocean and evolved ca. 90 million years ago[60], another key attribute for palaeo SST reconstructions.

In the future, it will be necessary to evaluate the EZ₂₅ SST proxy further through its measurement in marine sediments from a larger array of spatial and temporal settings in order to identify constraints, including any potential modification to the calibration or proxy error (ca. 2.8 °C in the previously published water column investigation[32]). Here, we employed a calibration of EZ₂₅ with in situ water column temperature[32] to derive past SSTs in sediments, but the extent to which this can be carried out, more

**Fig. 5 | Compilation of YD/Holocene SST and other proxy records from the Barents/Norwegian Sea. a** IP$_{25}$ record from core 2 (JM99-1200) located in the Andfjorden, northern Norway (Fig. 1b) showing the presence of seasonal sea ice during the YD cold stadial[47]; **b** SST reconstructions for core 2 (JM99-1200). The dark blue line corresponds to the continuous EZ$_{25}$-based SST record, while the orange and red lines represent partial SST records derived from alkenones[49] and a foraminiferal transfer function[48], respectively. The dramatic decline in EZ$_{25}$-based SSTs during the Younger Dryas cold stadial (ca. 12.9–11.9 cal kyr BP) is bracketed by substantially warmer SSTs during the Bølling–Allerød warm interval and the early Holocene Thermal Maximum. For comparison purposes, the previously published alkenone-based SST record for core 1 (MD95–2011[50];) is also shown (brown line). The apparent decline in SST during the YD is likely an artefact driven by the relatively high %C$_{37:4}$ content (light blue line), probably derived from certain Group Iii Isochrysidales[29]; **c** EZ$_{25}$ SST reconstruction for core 6 (JM09-KA11-GC) located in the Kveithola Trough (western Barents Sea; Fig. 1b); **d** EZ$_{25}$ SST reconstruction for core **7** (M23258; dark blue line) (western Barents Sea; Fig. 1b) along with previous alkenone-based SSTs for the same core (brown line[52,53]) and core 8 (SV04; orange line[54]). The apparently lower SSTs in core 7 (M23258) during the YD and B-A are due to the relatively high %C$_{37:4}$ alkenone (light blue line) derived from sea ice sources[29]; **e** EZ$_{25}$ SST reconstruction for core 11 (NP05-11-70GC) located in the Olga Basin (northern Barents Sea; Fig. 1b). In each case, the proxy standard error is indicated with a vertical error bar symbol using the same colour as the corresponding data plot.

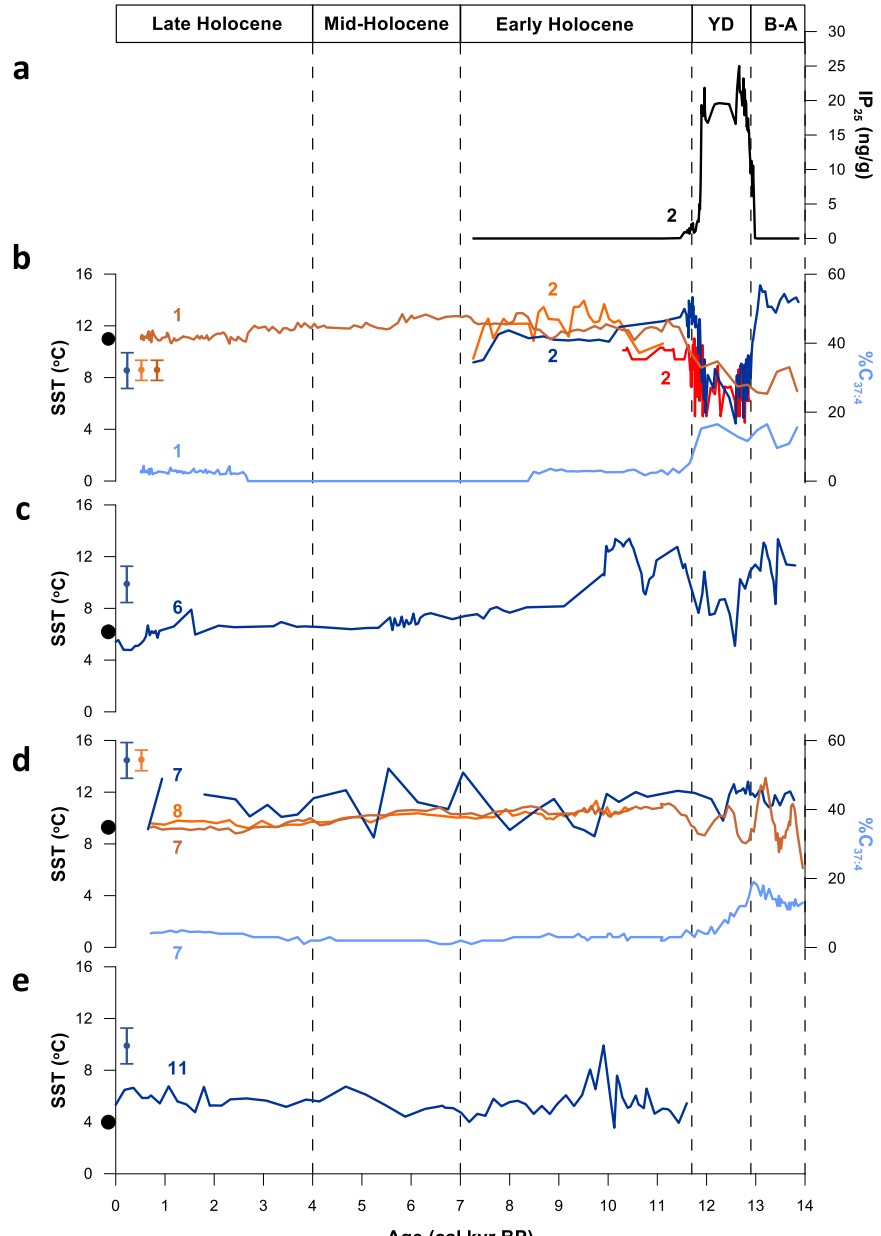

generally, will require a further assessment of EZ$_{25}$ in, for example, surface sediments, such that outcomes can be compared against recent SSTs measured by satellites or other observational records. However, analysis of surface sediments can be limited in some instances by the timeframes that they represent, especially in low accumulation rate regions such as the central Arctic Ocean, and concerns regarding biomarker preservation in old or inadequately stored material might also be important[61]. In contrast, many of the records presented here were conducted on radiometrically-dated short cores that had been stored at low temperature (< 4 °C). In fact, the importance of biomarker degradation, be it in the water column or in sediments, will also need investigation, as will the possible interconversion of HBIs III and IV (i.e. isomerisation). Although the EZ$_{25}$ proxy benefits to some extent from being a ratio-based method, any differential degradation of HBI III or HBI IV will result in a warm or cold bias in reconstructed SSTs, respectively. Further complications may arise if there are additional non-thermal factors that influence the relative amounts of HBIs III and IV or if their source diatom species composition changes over time. The latter may become important if species within genera other than *Rhizosolenia* and *Pleurosigma* are identified as producers of HBIs III and IV. Similarly, it will

be important to determine the water column depth characteristics of EZ$_{25}$ over a range of seasons to further pinpoint the temperatures it most accurately represents when measured in underlying sediments. It may be of additional interest to investigate whether EZ$_{25}$ can be employed for SST reconstructions at ocean temperatures higher than those reported here. For now, we suggest that EZ$_{25}$-based ocean temperatures likely reflect spring–summer SSTs since this is the main period of diatom growth and the data presented here are consistent with this. In some previous work, there have been suggestions that production of HBI III (and maybe HBI IV) reaches a maximum within the open waters of the retreating sea ice edge in late spring (for sea-ice covered regions, at least)[43], which potentially provides some seasonal constraint. In contrast, at least one known producer of HBIs III and IV (i.e., *Rhizosolenia*) can bloom between spring and autumn[32], which therefore also needs considering when interpreting EZ$_{25}$ SSTs. Finally, we re-iterate the importance of accurate identification and quantification of HBIs III and IV, something that has not always been evident in some previous studies where HBIs have been measured[62]. Further guidelines on particular experimental protocols required to achieve such accuracy have been reported in detail elsewhere[32,62,63].

**Fig. 6 | Proxy data from Fram Strait over the past 750 kyr and related information.** Proxy data from core 10 (ODP 910 A)[58] unless otherwise stated. **a** LR04 global stack ($\delta^{18}$O benthic foraminifera[70]) showing alternating marine isotope stages; **b** EZ$_{25}$ SST reconstruction; **c** Spring sea ice concentration (SpSIC (%)) based on IP$_{25}$ and HBI III[68]; **d** IP$_{25}$ concentration showing seasonal sea ice bioproductivity; **e** Brassicasterol concentration (mainly from phytoplankton bioproductivity); **f** Dinosterol concentration (mainly from dinoflagellate bioproductivity); **g** $\delta^{18}$O planktic foraminifera. The standard error of the EZ$_{25}$ SST proxy (2.8 °C) from an earlier water column calibration[32] is indicated by a vertical error bar symbol.

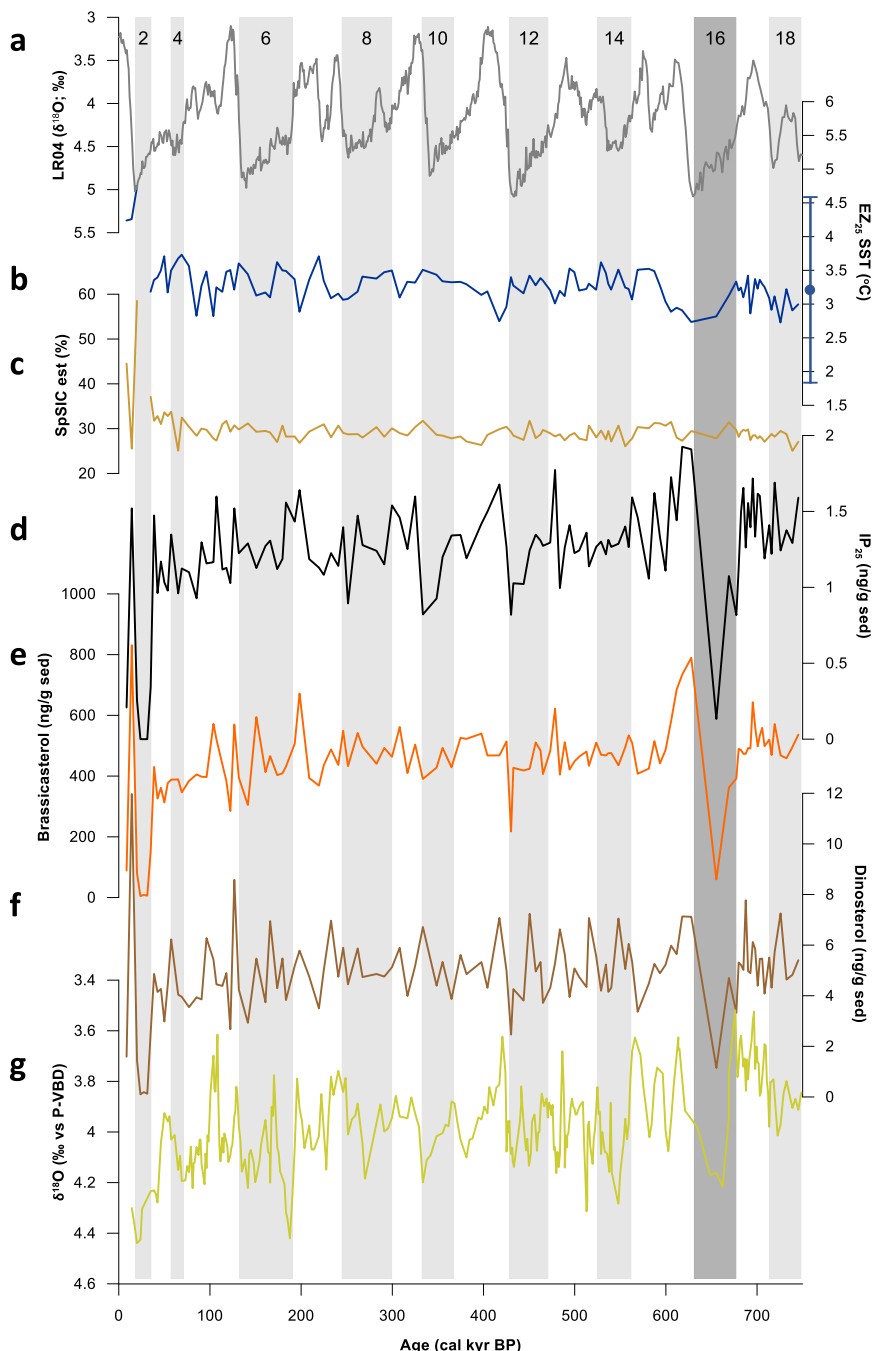

In summary, the variable distribution of two common lipid biomarker constituents of marine sediments, worldwide (i.e. HBIs III and IV), may provide a window of opportunity for assessing temporal and spatial changes to past surface ocean temperatures, including for the polar regions, where contemporary change is most evident.

## Methods

HBI biomarker data have either been reported previously, were obtained by re-analysis of gas chromatography–mass spectrometry (GC–MS) chromatograms obtained as part of these earlier studies, or determined from newly sampled sediment material. Sources of other literature biomarker data (i.e., alkenones) are cited in the main text. Previously reported HBI biomarker data were used for cores R248MC010 ($n = 24$), BASICC 1 ($n = 17$), BASICC 43 ($n = 19$), MSM5/5–712–1 ($n = 43$), JM99–1200 ($n = 114$), JM09–KA11–GC ($n = 116$) and NP05–11–70GC

($n = 67$)[64,65]. Some partial HBI biomarker data for ODP 1098 have been reported previously by other researchers but were not quantitative[57] so we obtained new data for this core ($n = 93$) and for cores M23258 ($n = 53$), BC 523 ($n = 14$) and BC 660 ($n = 14$). For these cores, HBI data were obtained from freeze dried sediment material according to the following methodology. After addition of an internal standard (9-octylheptadec-8-ene, 9-OHD, 0.10 µg) to weighed sediment samples, subsequent extraction (KOH/MeOH/H$_2$O) and partial purification (silica, hexane), HBI-containing fractions were analysed by GC–MS using an Agilent 7890 gas chromatograph equipped with a HP$_{5MS}$ fused-silica column (30 m; 0.25 µm film thickness; 0.25 mm internal diameter) coupled to an Agilent 5975 series mass spectrometric detector in total ion current (TIC) and selected ion monitoring (SIM) modes[63]. HBIs were identified based on their characteristic GC retention indices (RI$_{HP5MS}$ = 2044 and 2091 for HBI III and HBI IV, respectively) and mass spectra[43,66]. HBI

quantification was achieved by comparison of mass spectral responses of the molecular ion of HBIs III and IV (i.e. $m/z$ 346) in SIM mode with those of the internal standard (9-OHD, $m/z$ 350) and normalised according to their respective instrumental response factors[62], derived by analysis of mixtures of known concentration of 9-OHD and purified HBIs III and IV extracted from a culture of the diatom *Pleurosigma intermedium*[66]. Estimates of spring sea ice concentration (SpSIC (%)) for ODP 910 A were obtained from abundances of the sea ice biomarker $IP_{25}$[43,66] and HBI III according to a method described previously[67,68]. All $EZ_{25}$ SST and related data can be found in Supplementary Table 1 (https://doi.org/10.6084/m9.figshare.30630980). Chronologies for all Arctic cores and ODP 1098 have all been reported previously[35,39,47–58,64,65,69]. BC 523 and BC 660 were collected as part of cruises JR179 (2008) and JR257 (2012), respectively, aboard the RRS James Clark Ross and dated using $^{210}Pb$. Age models for both box cores are based on polynomial regressions ($R^2 = 0.99$) of the continuous rate of supply (CRS) model (Supplementary Table 2; (https://doi.org/10.6084/m9.figshare.30630980)). For BC 523, excess $^{210}Pb$ activity was measured on 13 downcore samples in the top 23 cm of the core using a J-shaped ultra-low background germanium well detection system at Durham University, UK, while for BC 660, excess $^{210}Pb$ activity was measured on 14 downcore samples in the top 15 cm using an extra low background co-axial High-Purity Germanium detector (GWL-195-15-XLB-AWT-S) at Laval University, Canada. Data from the Aqua satellite (NASA)[33] equipped with a Moderate Resolution Imaging Spectroradiometer (MODIS) was used to retrieve monthly SST (2003–2022), which were then combined to give seasonal and annual values. Sea ice concentration data were obtained from the National Snow and Ice Datacenter[59].

## Data availability
All data needed to evaluate the conclusions are present in the paper and/or Supplementary Tables 1 and 2. $EZ_{25}$-based SST and related data including excess $^{210}Pb$ and the continuous rate of supply (CRS) models (excess) for box cores 523 and 660 can be found at: https://doi.org/10.6084/m9.figshare.30630980.

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

## Acknowledgements

We thank the Natural Environment Research Council (UK; NE/X009416/1) and the Leverhulme Trust (UK; RPG-2015-439) for funding to support this research. This work was partly supported by the EU through its Horizon Europe funding scheme, project number 101118519 I2B. We also thank Patricia Cabedo-Sanz, Alba Navarro-Rodriguez, Sarah Berben and Katrin Schmidt for some data acquisition.

## Author contributions

S.T.B. conceptualised and coordinated the study, carried out the data analysis and wrote the initial draft. L.S., C.S.A., and D.K. carried out some of the data analysis. S.T.B., L.S., D.K., C.S.A K.H., and J.K. contributed to the discussion of data and production of the final version of the manuscript.

## Competing interests

The authors declare no competing interests.
