## [Transparent Peer Review file · Communications Earth & Environment]

Diatom lipids open window to past ocean temperatures in the polar regions

Corresponding Author: Professor Simon Belt

Version 0:

Decision Letter:

Dear Professor Belt,

Your manuscript titled "Diatom lipids open window to past ocean temperatures in the polar regions" has now been seen by 3 reviewers, and we include their comments at the end of this message. They find your work of interest, but some important points are raised. We are interested in the possibility of publishing your study in Communications Earth & Environment, but would like to consider your responses to these concerns and assess a revised manuscript before we make a final decision on publication.

When revising, please consider the main issues, including improving the clarity of the manuscript's structure, ensuring the presentation and consistency of figures, and strengthening the discussion of the interpretation and robustness of the EZ25 proxy.

We therefore invite you to revise and resubmit your manuscript, along with a point-by-point response that takes into account the points raised. Please highlight all changes in the manuscript text file.

Please submit your point-by-point responses as a separate file, distinct from your cover letter where you can add responses to the Editors' comments that you do not want to be made available to the reviewers. Word files are preferred. We recommend that any figures, tables or graphs that are included in the response to reviewers are also included in the main article or Supplementary Information.

Please use the following link to submit your revised manuscript, point-by-point response to the referees' comments (which should be in a separate document to any cover letter), a tracked-changes version of the manuscript (as a PDF file) and the completed checklist:

Link Redacted

We hope to receive your revised paper within six weeks; please let us know if you aren't able to submit it within this time so that we can discuss how best to proceed. If we don't hear from you, and the revision process takes significantly longer, we may close your file. In this event, we will still be happy to reconsider your paper at a later date, as long as nothing similar has been accepted for publication at Communications Earth & Environment or published elsewhere in the meantime.

Please do not hesitate to contact us if you have any questions or would like to discuss these revisions further. We look forward to seeing the revised manuscript and thank you for the opportunity to review your work.

Best regards,

Deborah Tangunan
Editorial Board Member
Communications Earth & Environment
orcid.org/0000-0002-1078-5767

Alice Drinkwater, PhD
Associate Editor
Communications Earth & Environment
Consulting Editor
Communications Sustainability

EDITORIAL POLICIES AND FORMATTING

- Behavioural and social science
- Ecological, evolutionary & environmental sciences
- Life sciences

Furthermore, please align your manuscript with our format requirements, which are summarized on the following checklist: <https://www.nature.com/documents/commsj-phys-style-formatting-checklist-article.pdf> Communications Earth & Environment formatting checklist

and also in our style and formatting guide <https://www.nature.com/documents/commsj-phys-style-formatting-guide-accept.pdf> Communications Earth & Environment formatting guide .

***** DATA:** Communications Earth & Environment endorses the principles of the Enabling FAIR data project (<http://www.copdess.org/enabling-fair-data-project/>). We ask authors to make the data that support their conclusions available in permanent, publically accessible data repositories. (Please contact the editor if you are unable to make your data available).

All Communications Earth & Environment manuscripts must include a section titled "Data Availability" at the end of the Methods section or main text (if no Methods). More information on this policy, is available at <http://www.nature.com/authors/policies/data/data-availability-statements-data-citations.pdf>

If a community resource is unavailable, data can be submitted to generalist repositories such as <https://figshare.com/> or <http://datadryad.org/> Dryad Digital Repository. Please provide a unique identifier for the data (for example a DOI or a permanent URL) in the data availability statement, if possible. If the repository does not provide identifiers, we encourage authors to supply the search terms that will return the data. For data that have been obtained from publically available sources, please provide a URL and the specific data product name in the data availability statement. Data with a DOI should be further cited in the methods reference section.

REVIEWER COMMENTS:

Reviewer #1 (Remarks to the Author):

Review of Belt et al. for Nature Communications Earth and Environment
Joseph B. Novak

Summary

Belt et al. present multiple new sea surface temperature (SST) records derived from the EZ25 proxy, which is based upon highly branched isoprenoid (HBI) diatom biomarkers in sediments from the Arctic and Antarctic. The purpose of this study is to demonstrate the utility of the EZ25 SST proxy for constructing paleotemperature timeseries in polar regions. This is particularly important to the paleoclimate science community, as SSTs in polar regions are highly challenging to reconstruct with other methods. Furthermore, climate model simulations of past and future climate states suggest that polar temperatures are highly sensitive to changes in greenhouse gas concentrations, underscoring the need to constrain the ability of climate models to accurately emulate polar conditions.

Belt et al. demonstrate that EZ25 SST estimates conform with expected patterns of change in recent sediments and show the occurrence of the HBI biomarkers in strata extending to 750 thousand years before present. However, the 750-thousand-year timeseries of EZ25 SSTs shown in their Figure 6 substantially differs from the pattern expected from global climate changes during this interval, a finding which is not immediately clear from the construction of the figure or the manuscript text.

Overall, the findings presented here suggest that EZ25 has potential as an SST proxy in a region that has long been challenging for paleoclimatologists. I look forward to the publication of this work once my comments are addressed.

Major Comments

Abstract: from my initial read of the manuscript, my impression is that the intent is to address the following questions.

- 1). Does the EZ25 index qualitatively/quantitatively reproduce expected sea surface temperature trends in polar regions?
- 2). Is it possible to apply EZ25 to sedimentary strata where other SST biomarker proxies are absent or unreliable?

I think a weakness of this manuscript is that the abstract reads as though this is a calibration paper rather than a demonstration of the application of this proxy system designed to address the above questions. I got this impression from the abstract text discussing the linearity of the EZ25-SST relationship (Lines 20-26). Instead, I suggest emphasizing that a major strength of this proxy system is that the diatom-sourced HBI molecules are present in both surficial sediments and older strata where alkenones are largely absent. I think it would also be useful to state the extent to which the EZ25 SST reconstructions presented here conform with your expectations on different timescales (and to state in the abstract what those expectations are based upon).

L76–127 “SST records over recent centuries”: I struggled to glean a clear message from these three paragraphs. The results are thoroughly described, but what am I supposed to understand about the proxy system from these data? Better topic sentences and concluding sentences in each paragraph would be useful for guiding the reader to the points you are trying to make. For example, the paragraph starting at Line 76 may be easier to digest if it started with the following sentences:

“We analyzed HBIs in a collection of Arctic and Antarctic sediment cores to demonstrate the utility of the EZ25 proxy in different oceanographic settings and geologic timescales. SSTs estimated from the EZ25 index in short cores from the Arctic reflect temperature change through time as constrained from satellite observations and other geologic SST proxy datasets (Fig. 1c). For example...”

L130–133: This topic sentence does not really do your results justice. I think the points you are trying to convey are that (1) where there are other proxy data available, EZ25 agrees with those data and (2) the HBIs that comprise the EZ25 index are present in strata barren of other biomarkers used for SST estimation. I suggest reframing this topic sentence to set the reader up to expect a description of those observations.

Figure 1: I think this should be broken up into two figures - one figure with the molecular structures and map and another with the temperature proxy record time series. I am also somewhat confused by why panel c is repeated between figure 1 and figure 2. Lastly, the color scale in this map is not colorblind friendly. I suggest using red-blue to show SSTs as this is a much easier contrast for colorblind people than the rainbow scale.

Figure 4: The panels need to be more clearly labelled so that it is easier to understand which cores these data are coming from (and where those cores are). I found myself getting confused while flipping between the text describing this figure and the figure itself for this reason. Also, either the green or the red line needs to be a different color for colorblind accessibility.

Figure 6 and L197–201: The SST estimates appear to poorly reflect what I would intuitively expect for a glacial-interglacial SST pattern (that is, warmer temperatures during interglacial periods and colder temperatures during interglacial periods). This is not necessarily a major issue for the proxy, but the text and figure should present these findings more clearly. Specifically, the y-axis in panel a should be inverted, which is typical practice when plotting $\delta^{18}\text{O}$ data. This would help with the visual comparison of the SST estimates and the $\delta^{18}\text{O}$ data since the larger $\delta^{18}\text{O}$ values indicate colder and higher ice volume conditions.

Minor Comments

L17: "barometer" is a potentially confusing word here since EZ25 is an SST proxy rather than a CO2 proxy. I suggest using "indicator" instead.

L18–20: The commas on either side of "accurately" are not necessary.

L35: "boundary parameter" has a very specific meaning in computer science that I suspect is not what is intended here. I think it may be more appropriate to say "key variable" or something similar.

L47: probably best to specify "isoprenoid glycerol dialkyl glycerol tetraethers."

L43–49: This sentence would probably read better if it was broken up into two shorter sentences.

L53–61: While I know that these statements are true, they do need to be properly referenced.

L65: "in situ" should be italicized here and elsewhere.

L81 & L85: The second open parentheses in both places is not needed, just write "from" and then the years.

L88: should specify which figure panel in the callout here (e.g., Fig. 1b) and elsewhere.

L104: I suggest adding "(cf., ref.21)" here. Reference 21 being Wang et al. (2021) in Nat. Comms.

Figure 2: the legend for panels a and b are flipped. In panel b, the red or green line needs to be made a different color as the contrast is indistinguishable to red-green colorblind people. Panel a would probably benefit from being split into two panels: one that shows the SST trends and another that shows the %C37:4 alkenone timeseries.

Figure 5: Either the green or the red line needs to be a different color for colorblind accessibility. The panel labels here are reversed relative to all the other figures. Please make this consistent as it makes reading the figure legend unintuitive.

Reviewer #2 (Remarks to the Author):

Belt et al apply the newly developed EZ25-SST proxy to biomarker records from sediment core in both polar regions, which have decadal, centennial and millennial to orbital timescales. The goal of the manuscript is to show the feasibility of applying this new proxy in regions where existing temperature-sensitive proxies do not perform very well or are limited due to calibration (mainly) issues. The result of this study is important, since in a few cases it compares the downcore EZ25-SST records with other temperature and surface ocean conditions proxies. In general, appropriate methods are used, although I have a few suggestions for the authors to more clearly show the results, and to improve some of the discussion and the figures.

As a general comment, I think the authors should include the error bars of the EZ25-SST proxy and of the other temperature proxies in the plots. Some parts of the discussion section could also be more clearly written; detailed suggestions are found below. Please revise the figures (colors, legends), call out of figures should include panels in the text. Captions are sometimes not clear or are missing proxy data (detailed suggestions below).

Discussion is sometimes difficult to follow since it jumps between figures (and panels which are not easy to identify since they are not included in the figure call out).

Overall, the data shown here support the conclusions. I have also suggestions regarding the composition of the figures. I also noted that there is not statement about data availability, and I strongly recommend that the new EZ25 is made publicly available once this study is accepted. With some moderate revision of the text and figures, I think this manuscript would represent a strong and useful contribution to the paleo community.

Other comments

L. 17. I understand that the authors use "barometer" referring to sea surface temperature as a metaphor, but in my opinion, science needs to be factual and would refrain from using here a word that describes a device which is used for measuring air pressure, not temperature.

L. 46. Conte's global dataset has a temperature range from -1 to 30°C. Even if that limitation existed in the original dataset and regression, newer calibrations, such as those by Tierney and Tiengly 2018 and Novak et al. 2022, have improved the UK'37 calibrations, and did not find neither limitation for reconstructing temperatures below 8°C. These calibrations perform adequately in the cold Southern Ocean, suggesting that this is not a thermal limitation of the proxy but rather the presence of confounding influences in the Arctic region, and there could be larger regression uncertainties in the colder end due to limited number of core tops available between 0–5°C (Novak et al. 2022).

L. 48. Similarly to the previous comment, I think the authors should acknowledge other views and more recent papers on the "cold-water limitation" of the GDGTs. There are a few more recent studies after Kim's calibration that have proposed new indices and develop new approaches to use isoGDGT-based palaeothermometry also in cold regions (e.g., Dunkley Jones et al., 2020; Tierney and Tingley, 2014). Particularly, Ishii et al (in review) and Park et al (2019) have proposed new indices to overcome this cold-temperature limitation.

L. 62. Reference needed. For example, Harada et al. (2003)

It would be easier if the authors could include the panel letter when they call the figures in the text (e.g. at L. 88; Fig. 1c).

L. 141. It should be figure 4a, not 3.

L. 142. The authors mention presence of sea biomarker IP25 in figure 4a, but the caption corresponding to that figure does not include anything about IP25. Please revise. Besides, the text in L. 145 and the legend of figure 4a mention alkenones and planktonic foraminifera-based SST reconstructions (orange and red lines) but I only see a red line, which according to the legend is the alkenone-derived SST. What the blue line in Figure 4a corresponds to? Finally, I do not see any data from MD95–2011 in that panel, although I think it should be expected based on what the text (L. 146-153).

The text corresponding to ODP site 1098 and Figure 4b (starting at L. 182) should appear before the discussion of JM09KA11-GC, whose records appear in figure 5. Caption figure 1. Information corresponding to panels c and d are swapped in the caption. These two panels would be easier to follow if they included a legend next to the panels, as done in figure 2. In the same caption, I think the authors should change SIMMAX planktic foraminifer- transfer function, or similar, because not everyone links SIMMAX with a planktonic foraminifera-based temperature.

L. 182. I wonder if it makes sense to compare the new EZ25-based SSTs at 1098 with previously published TEX-86, which are considered as not very meaningful according to some of the limitations in the calibration that existed when the original record was published. Moreover, there are substantial differences in the EZ25 and TEX86-based SSTs during the early Holocene, that should be explained.

L. 189. I do not find the comparison between LR04 and the new EZ25-SST record from ODP site 910 very meaningful. Rather than figure 6, I think it would be more interesting to compare the new SST record with the oxygen isotope planktonic foraminifera record shown in the Supplementary figure. Regarding the coldest temperatures, I do not think that there is much difference between SST during MIS 16 and 18. I think it would be interesting having some discussion about the trends of the new SST record, and why there is not always glacial-interglacial variability. Indeed, often higher SST values are recorded during glacial intervals (MIS 14, 12 and 4), or viceversa, why during interglacial intervals, such as early MIS 11, we can see a cooling rather than a warming. Moreover, I would find important to discuss the differences between the planktonic foraminifera $\delta^{18}O$ record and the new SST, since some of the heaviest isotope values (e.g. MIS 6 or MIS 14) (OK, it is true that could be due to either colder temperatures or very high salinity, or a combination of both factors) do not correspond with very low temperatures. I would also find interesting the discussion of IP25 record and the SST, to see if there are potential/expected biases due to the presence of sea ice which could have an effect on the diatom assemblages.

Comments on the figures

Is panel Fig. 1c necessary since it is shown again in Figure 2 (plus three additional proxy curves)?

In figure 4a, the use of green and red in the proxy curves will make things difficult to color-blind people. Please use other colors. Same for figure 5b with brown, red and green.

Figure 6. I am not sure if it is a formal convention in isotope geochemistry in paleoceanography, but I would ask the authors to invert the axis of the LR04.

References

- Dunkley Jones, Tom, et al. "OPTiMAL: A new machine learning approach for GDGT-based palaeothermometry." *Climate of the Past* 16.6 (2020): 2599-2617.
- Harada, Naomi, et al. "Characteristics of alkenones synthesized by a bloom of *Emiliana huxleyi* in the Bering Sea." *Geochimica et Cosmochimica Acta* 67.8 (2003): 1507-1519.
- Ishii et al. "New isoprenoid GDGT index as a water mass and temperature proxy in the Southern Ocean." *EGUsphere* 2025 (2025): 1-24.
- Kaiser, Jérôme, et al. "Changes in long chain alkenone distributions and Isochrysidales groups along the Baltic Sea salinity gradient." *Organic Geochemistry* 127 (2019): 92-103.
- Novak, Joseph, et al. "U38MEK' Expands the linear dynamic range of the alkenone sea surface temperature proxy." *Geochimica et Cosmochimica Acta* 328 (2022): 207-220.
- Park et al. "Seasonality of archaeal lipid flux and GDGT-based thermometry in sinking particles of high-latitude oceans: Fram Strait (79 N) and Antarctic Polar Front (50 S)." *Biogeosciences* 16.11 (2019): 2247-2268.
- Tierney, J. E., & Tingley, M. P. (2018). BAYSPLINE: A new calibration for the alkenone paleothermometer. *Paleoceanography and Paleoclimatology*, 33, 281–301. <https://doi.org/10.1002/2017PA003201>
- Tierney, Jessica E., and Martin P. Tingley. "A Bayesian, spatially-varying calibration model for the TEX86 proxy." *Geochimica et Cosmochimica Acta* 127 (2014): 83-106.

Reviewer #3 (Remarks to the Author):

This manuscript presents downcore data for a new biomarker-based temperature proxy for use in cold water settings. It's great to see new proxies being developed in these challenging settings, and EZ25 has the potential to be a really important

component of polar paleoclimate work. It's encouraging to see that the temperatures reconstructed are for the most part realistic and align with other data. However, I have several concerns about the paper as it's currently presented.

Firstly, as detailed more below, there are numerous errors, especially throughout the figures where some careful revising and checking is required.

Secondly, I understand that an initial paper has just been published presenting this proxy in Organic Geochemistry, but this manuscript currently reads as if it is just an extension of that, rather than a new paper published in a different journal. I found myself needing to repeatedly go back to the Organic Geochemistry paper for clarification or to see if questions I had were answered in there. Some degree of that is always expected when reviewing a manuscript, but in this case I think more work is required to better ensure this paper stands alone. In particular, the results and site specific interpretations of the down core records are described, but the discussion as to what all these results mean for the proxy is limited. What do these results suggest for the impact of seasonality on the proxy? Or other factors that can often confound other proxies, like changes in species present through time, salinity, nutrients, surface or subsurface signal etc? The outlook section is very brief, what sort of next steps would help to improve the proxy or determine aspects like the upper temperature limit, or how far back in time the proxy could go to, or if species other *Rhizosolenia* could be producing the constituent HBIs?

This sounds like a proxy with a lot of potential, and it's important to see a more thorough discussion and presentation of ideas for refinement following the description of results. Some line/figure comments below.

Line 43: Sentence starting 'For', suggest for readability this sentence is split into two as it's quite hard to follow. While I definitely agree there are challenges with applying a GDGT-based paleothermometer at lower temperatures, the description here an oversimplification that doesn't acknowledge there's been a lot of work on better calibrating polar GDGTs. For instance spatially varying or regional specific calibrations, or OH-GDGT-based proxies (i.e. Fietz et al., 2016 <https://doi.org/10.1016/j.orggeochem.2016.10.003>, Varma et al., 2024, <https://doi.org/10.1016/j.gca.2023.12.019>). I think your point can still hold by acknowledging that a lot of work has been done on polar temperatures proxies, but there are still challenges to the current methods we try to use at high latitudes, and that any new proxy that can better reconstruct polar temperatures is very welcome.

Line 56: Correct format of UK37.

Line 147: Remove 'in' from before JM99-1200.

Lines 197-201: See comments about Figure 7.

Line 365: What supplementary information is this referring to? Supplementary Table 1 doesn't appear in the review document? Will the new data that is referred to in this paper be included in a supplementary table too?

Methods: This jumps around a bit and it would help if it was more structured i.e. shift to the top what core data has been previously reported, and what is new. With the exception of the age models for the box cores developed here, where have the age models been reported for the previously studied cores, and cores that other new data has been analysed from?

Figures all need to be carefully checked.

Figure 1: c) and d) are labelled around the wrong way. Is the black dot on c) the modern temperature for core 12?

Figure 2: a) and b) are described the wrong way around in the caption. The panel looks to be the same as panel b) from Figure 1. Suggest that these figures are reworked so the same panel isn't shown twice, as well as ensuring the captions are correct.

Figure 3 caption describes core 13 as in panel a) but the figure has it numbered on panel b).

Figure 4 describes an orange line in the caption but I believe this in blue in the figure.

Figure 5 has swapped the order of a)-e) from bottom to top, compared to the other figures where a) is at the top. Panel b) describes an orange line instead of blue. d) looks to have a dark blue line rather than purple?

Figures 6 and 7: Convention is that the LR04 stack is shown with higher values on the bottom i.e. with the y axis reversed. In particular on Figure 7, having this axis reversed makes it quite hard to read the figure, especially as the $\delta^{18}O$ from the core itself is shown the opposite. While the reconstructed temperature appear appropriate, there does not seem to be a particularly clear relationship between warmer temperatures during interglacials and cooler in glacials. The wording of lines 197 to 201 in the text glosses over this and instead only highlights cool temperatures in MIS 16 and warm 'post LGM', when there's also cool data at the height of MIS 11, and the warmest sample actually looks to be in the LGM. I'd prefer this was more clearly discussed, and instead used as a good opportunity to discuss some potential reasons for this variability and lack of clear glacial/interglacial pattern (i.e. could this be to do with the site-specific environment or is there some interglacial/glacial change in something like the diatom HBI producer that could impact how the proxy works).

** Visit Nature Portfolio's author and referees' website at www.nature.com/authors for information about policies, services and author benefits**

Communications Earth & Environment is committed to improving transparency in authorship. As part of our efforts in this direction, we are now requesting that all authors identified as 'corresponding author' create and link their Open Researcher and Contributor Identifier (ORCID) with their account on the Manuscript Tracking System prior to acceptance. ORCID helps the scientific community achieve unambiguous attribution of all scholarly contributions. You can create and link your ORCID from the home page of the Manuscript Tracking System by clicking on 'Modify my Springer Nature account' and following the instructions in the link below. Please also inform all co-authors that they can add their ORCID to their accounts and that they must do so prior to acceptance.

Version 1:

Decision Letter:

Dear Professor Belt,

Your manuscript titled "Diatom lipids open window to past ocean temperatures in the polar regions" has now been seen by our reviewers, whose comments appear below. In light of their advice we are delighted to say that we are happy, in principle, to publish a suitably revised version in Communications Earth & Environment.

We therefore invite you to revise your paper one last time to address the remaining concerns of our reviewers. At the same time we ask that you edit your manuscript to comply with our format requirements and to maximise the accessibility and therefore the impact of your work.

EDITORIAL REQUESTS:

****Please take care to match our formatting and policy requirements. We will check revised manuscript and return manuscripts that do not comply. Such requests will lead to delays. ****

SUBMISSION INFORMATION:

OPEN ACCESS:

Communications Earth & Environment is a fully open access journal. Articles are made freely accessible on publication. For further information about article processing charges, open access funding, and advice and support from Nature Portfolio, please visit <https://www.nature.com/commsenv/open-access>

Link Redacted

Best regards,

Alice Drinkwater, PhD
Associate Editor
Communications Earth & Environment
Consulting Editor
Communications Sustainability

REVIEWERS' COMMENTS:

Reviewer #1 (Remarks to the Author):

2nd Review of Belt et al. for Nature Communications Earth and Environment
Joseph B. Novak

Summary

The revised version of the manuscript presented by Belt et al. addresses my major concerns. I look forward to seeing this work published.

Minor Comments

L6: Typo here.

L305: The title of reference 12 is incorrect.

Reviewer #2 (Remarks to the Author):

This is the second time I have seen this manuscript. Overall, I think that it is much improved but could still use some editing. The authors present an interesting interpretation of their data in the context of the EZ25 data. I am satisfied with how the authors have dealt with reviewer comments. I only have a couple of comments:

I will think that the authors should include standard error in the figures, even if this error does not include the error propagation (analytic errors). It is meaningful to see whether some of the changes in the EZ25 data are meaningful (larger than the calibration error) or not. This is important for example in ODP Site 910, where the authors acknowledge that the SST changes are too small to be taken into account.

L. 74: "makes selection of the most appropriate, problematic". I do not really agree that selection is problematic, and the data can be reconstructed using different calibrations...

Reviewer #3 (Remarks to the Author):

Thank you to the authors for revising the manuscript and correcting the figure errors. My comments from the initial review have been mostly addressed and the manuscript has been improved. Including topic sentences in the results and discussion has improved the readability of the section. The outlook section has also been expanded and now includes a more thorough discussion of potential confounding factors for the proxy as well as future research directions. A couple of small comments.

Lines 44-49: Suggest this sentence is split into two for readability.

Lines 71-79: Similar to a comment by another reviewer, I think adding a sentence in here about the standard error of the proxy in the water column (and why you have not applied it to the sediment data) is important context for interpreting the results. It would be good to still see this section expanded a bit more to better introduce the proxy, i.e. adding a more specific upfront comment on the potential diatom source for this proxy, rather than waiting to discuss this in the outlook section.

Line 280: I recommend a conclusion statement is added to the end of the section as a final summary, as the newly added text finishes rather abruptly. I appreciate the addition of this outlook information, it sets the scene for future application of the proxy more clearly.

** Visit Nature Portfolio's author and referees' website at www.nature.com/authors for information about policies, services and author benefits**

Responses to Reviewers (**in bold**)

We would like to thank all three reviewers for their supportive comments regarding the potential of EZ₂₅ as a new SST proxy for the polar regions. We also apologise for the errors associated with Figure legends and labelling.

Reviewer #1 (Remarks to the Author):

Review of Belt et al. for Nature Communications Earth and Environment
Joseph B. Novak

Summary

Belt et al. present multiple new sea surface temperature (SST) records derived from the EZ₂₅ proxy, which is based upon highly branched isoprenoid (HBI) diatom biomarkers in sediments from the Arctic and Antarctic. The purpose of this study is to demonstrate the utility of the EZ₂₅ SST proxy for constructing paleotemperature timeseries in polar regions. This is particularly important to the paleoclimate science community, as SSTs in polar regions are highly challenging to reconstruct with other methods. Furthermore, climate model simulations of past and future climate states suggest that polar temperatures are highly sensitive to changes in greenhouse gas concentrations, underscoring the need to constrain the ability of climate models to accurately emulate polar conditions.

Belt et al. demonstrate that EZ₂₅ SST estimates conform with expected patterns of change in recent sediments and show the occurrence of the HBI biomarkers in strata extending to 750 thousand years before present. However, the 750-thousand-year timeseries of EZ₂₅ SSTs shown in their Figure 6 substantially differs from the pattern expected from global climate changes during this interval, a finding which is not immediately clear from the construction of the figure or the manuscript text.

Overall, the findings presented here suggest that EZ₂₅ has potential as an SST proxy in a region that has long been challenging for paleoclimatologists. I look forward to the publication of this work once my comments are addressed.

Major Comments

Abstract: from my initial read of the manuscript, my impression is that the intent is to address the following questions.

- 1). Does the EZ₂₅ index qualitatively/quantitatively reproduce expected sea surface temperature trends in polar regions?
- 2). Is it possible to apply EZ₂₅ to sedimentary strata where other SST biomarker proxies are absent or unreliable?

I think a weakness of this manuscript is that the abstract reads as though this is a calibration paper rather than a demonstration of the application of this proxy system designed to address the above questions. I got this impression from the abstract text discussing the linearity of the EZ₂₅-SST relationship (Lines 20-26). Instead, I suggest emphasizing that a major strength of this proxy system is that the diatom-sourced HBI molecules are present in both surficial

sediments and older strata where alkenones are largely absent. I think it would also be useful to state the extent to which the EZ25 SST reconstructions presented here conform with your expectations on different timescales (and to state in the abstract what those expectations are based upon).

We are not sure what the reviewer is referring to in the Abstract since the linearity of the EZ25-SST relationship is not described. In any case, we have focussed the Abstract on the key findings, which also takes into account the strict word limit (150). This has involved cutting the original by ca. 50 words.

L76–127 “SST records over recent centuries”: I struggled to glean a clear message from these three paragraphs. The results are thoroughly described, but what am I supposed to understand about the proxy system from these data? Better topic sentences and concluding sentences in each paragraph would be useful for guiding the reader to the points you are trying to make. For example, the paragraph starting at Line 76 may be easier to digest if it started with the following sentences:

“We analyzed HBIs in a collection of Arctic and Antarctic sediment cores to demonstrate the utility of the EZ25 proxy in different oceanographic settings and geologic timescales. SSTs estimated from the EZ25 index in short cores from the Arctic reflect temperature change through time as constrained from satellite observations and other geologic SST proxy datasets (Fig. 1c). For example...”

This is helpful piece of feedback and we have added new text at the beginning and end of each of the two main results sections to introduce/summarise the data.

L130–133: This topic sentence does not really do your results justice. I think the points you are trying to convey are that (1) where there are other proxy data available, EZ25 agrees with those data and (2) the HBIs that comprise the EZ25 index are present in strata barren of other biomarkers used for SST estimation. I suggest reframing this topic sentence to set the reader up to expect a description of those observations.

See previous response

Figure 1: I think this should be broken up into two figures - one figure with the molecular structures and map and another with the temperature proxy record time series. I am also somewhat confused by why panel c is repeated between figure 1 and figure 2. Lastly, the color scale in this map is not colorblind friendly. I suggest using red-blue to show SSTs as this is a much easier contrast for colorblind people than the rainbow scale.

We prefer to keep as one figure since the dataplots with numbered cores can be navigated alongside the map (eliminates the problem of map/data figures being on separate pages in the printed form)

We have re-plotted the data with the suggested red/blue colour palette.

Figure 4: The panels need to be more clearly labelled so that it is easier to understand which cores these data are coming from (and where those cores are). I found myself getting confused while flipping between the text describing this figure and the figure itself for this reason. Also, either the green or the red line needs to be a different color for colorblind accessibility.

We have now added the core numbers to the individual datalines (as per Figure 1) to assist the navigation. We have changed all figure colours to remove the red/green combination

Figure 6 and L197–201: The SST estimates appear to poorly reflect what I would intuitively expect for a glacial-interglacial SST pattern (that is, warmer temperatures during interglacial periods and colder temperatures during interglacial periods). This is not necessarily a major issue for the proxy, but the text and figure should present these findings more clearly. Specifically, the y-axis in panel a should be inverted, which is typical practice when plotting $\delta^{18}\text{O}$ data. This would help with the visual comparison of the SST estimates and the $\delta^{18}\text{O}$ data since the larger $\delta^{18}\text{O}$ values indicate colder and higher ice volume conditions.

We have modified the text and replaced Fig 6 with an updated version of Supp Fig 1. We have added in new sea ice concentration data which supports the consistency in SST at the core site.

Minor Comments

L17: “barometer” is a potentially confusing word here since EZ25 is an SST proxy rather than a CO₂ proxy. I suggest using “indicator” instead.

Changes, as requested

L18–20: The commas on either side of “accurately” are not necessary.

Changed, as requested

L35: “boundary parameter” has a very specific meaning in computer science that I suspect is not what is intended here. I think it may be more appropriate to say “key variable” or something similar.

Changed, as requested

L47: probably best to specify “isoprenoid glycerol dialkyl glycerol tetraethers.”

Changed, as requested

L43–49: This sentence would probably read better if it was broken up into two shorter sentences.

Changed, as requested

L53–61: While I know that these statements are true, they do need to be properly referenced.

Some additional key references have been added.

L65: “in situ” should be italicized here and elsewhere.

According to Springer Nature guidelines, in situ should not be italicised.

L81 & L85: The second open parentheses in both places is not needed, just write “from” and then the years.

Changed, as requested

L88: should specify which figure panel in the callout here (e.g., Fig. 1b) and elsewhere.

We have updated all figure numbers and added panel labels where previously missing.

L104: I suggest adding “(cf., ref.21)” here. Reference 21 being Wang et al. (2021) in Nat. Comms.

Reference 21 has now been cited here and updated (30)

Figure 2: the legend for panels a and b are flipped.

Corrected

In panel b, the red or green line needs to be made a different color as the contrast is indistinguishable to red-green colorblind people.

Changed as requested

Panel a would probably benefit from being split into two panels: one that shows the SST trends and another that shows the $\delta^{13}C_{37:4}$ alkenone timeseries.

Changed as requested

Figure 5: Either the green or the red line needs to be a different color for colorblind accessibility. The panel labels here are reversed relative to all the other figures. Please make this consistent as it makes reading the figure legend unintuitive.

Changed as requested

Reviewer #2 (Remarks to the Author):

Belt et al apply the newly developed EZ25-SST proxy to biomarker records from sediment core in both polar regions, which have decadal, centennial and millennial to orbital timescales. The goal of the manuscript is to show the feasibility of applying this new proxy in

regions where existing temperature-sensitive proxies do not perform very well or are limited due to calibration (mainly) issues. The result of this study is important, since in a few cases it compares the downcore EZ25-SST records with other temperature and surface ocean conditions proxies. In general, appropriate methods are used, although I have a few suggestions for the authors to more clearly show the results, and to improve some of the discussion and the figures.

As a general comment, I think the authors should include the error bars of the EZ25-SST proxy and of the other temperature proxies in the plots. Some parts of the discussion section could also be more clearly written; detailed suggestions are found below. Please revise the figures (colors, legends), call out of figures should include panels in the text. Captions are sometimes not clear or are missing proxy data (detailed suggestions below).

Discussion is sometimes difficult to follow since it jumps between figures (and panels which are not easy to identify since they are not included in the figure call out).

Overall, the data shown here support the conclusions. I have also suggestions regarding the composition of the figures. I also noted that there is not statement about data availability, and I strongly recommend that the new EZ25 is made publicly available once this study is accepted. With some moderate revision of the text and figures, I think this manuscript would represent a strong and useful contribution to the paleo community.

We address the above against the specific points below. Note that at this stage, although there is a standard error for EZ25 from the water column calibration (Belt et al., 2025), this may not translate to sediments and there is need of further measurements to establish random, systematic and analytical errors. To include error bars at this point would be mis-leading in our view. We do, however, refer to the SE within the text.

Other comments

L. 17. I understand that the authors use “barometer” referring to sea surface temperature as a metaphor, but in my opinion, science needs to be factual and would refrain from using here a word that describes a device which is used for measuring air pressure, not temperature.

Changed, as requested and as per Reviewer 1

L. 46. Conte’s global dataset has a temperature range from -1 to 30°C. Even if that limitation existed in the original dataset and regression, newer calibrations, such as those by Tierney and Tiengly 2018 and Novak et al. 2022, have improved the Uk’37 calibrations, and did not find neither llimitationfor reconstructing temperatures below 8°C. These calibrations perform adequately in the cold Southern Ocean, suggesting that this is not a thermal limitation of the proxy but rather the presence of confounding influences in the Arctic region, and there could be larger regression uncertainties in the colder end due to limited number of core tops available between 0–5°C (Novak et al. 2022).

We have revised the Introduction accordingly and cited some further papers, as suggested

L. 48. Similarly to the previous comment, I think the authors should acknowledge other views and more recent papers on the “cold-water limitation” of the GDGTs. There are a few more recent studies after Kim’s calibration that have proposed new indices and develop new approaches to use isoGDGT-based palaeothermometry also in cold regions (e.g., Dunkley Jones et al., 2020; Tierney and Tingley, 2014). Particularly, Ishii et al (in review) and Park et al (2019) have proposed new indices to overcome this cold-temperature limitation.

We have revised the Introduction accordingly and cited some further papers, as suggested

L. 62. Reference needed. For example, Harada et al. (2003)

This has been added, as requested

It would be easier if the authors could include the panel letter when they call the figures in the text (e.g. at L. 88; Fig. 1c).

Changed as per Reviewer 1

L. 141. It should be figure 4a, not 3.

Noted and changed

L. 142. The authors mention presence of sea biomarker IP25 in figure 4a, but the caption corresponding to that figure does not include anything about IP25. Please revise.

This should have been Figure 5a and has been updated.

Besides, the text in L. 145 and the legend of figure 4a mention alkenones and planktonic foraminifera-based SST reconstructions (orange and red lines) but I only see a red line, which according to the legend is the alkenone-derived SST. What the blue line in Figure 4a corresponds to?

We have updated all colours within the figures and checked against legends

Finally, I do not see any data from MD95–2011 in that panel, although I think it should be expected based on what the text (L. 146-153).

These data are in Figure 5 – now updated in the text.

The text corresponding to ODP site 1098 and Figure 4b (starting at L. 182) should appear before the discussion of JM09KA11-GC, whose records appear in figure 5.

We see this point, but we believe that a continuous presentation of the northern hemisphere records makes more sense, followed by a switch to the Antarctic. The decision to group the two records in Figure 4 was based on highlighting the post-glacial SST records for key sites, one from each of the N/S hemispheres.

Caption figure 1. Information corresponding to panels c and d are swapped in the caption.

We have switched the figures to align with the legend

These two panels would be easier to follow if they included a legend next to the panels, as done in figure 2. In the same caption, I think the authors should change SIMMAX planktic foraminifer- transfer function, or similar, because not everyone links SIMMAX with a planktonic foraminifera-based temperature.

The legend text and on the figure have been updated, as requested

L. 182. I wonder if it makes sense to compare the new EZ25-based SSTs at 1098 with previously published TEX-86, which are considered as not very meaningful according to some of the limitations in the calibration that existed when the original record was published. Moreover, there are substantial differences in the EZ25 and TEX86-based SSTs during the early Holocene, that should be explained.

We have added a comment on this – the small difference is either due to the calibration issue (TEX86) or possibly seasonality differences between proxies

L. 189. I do not find the comparison between LR04 and the new EZ25-SST record from ODP site 910 very meaningful. Rather than figure 6, I think it would be more interesting to compare the new SST record with the oxygen isotope planktonic foraminifera record shown in the Supplementary figure. Regarding the coldest temperatures, I do not think that there is much difference between SST during MIS 16 and 18. I think it would be interesting having some discussion about the trends of the new SST record, and why there is not always glacial-interglacial variability. Indeed, often higher SST values are recorded during glacial intervals (MIS 14, 12 and 4), or viceversa, why during interglacial intervals, such as early MIS 11, we can see a cooling rather than a warming. Moreover, I would find important to discuss the differences between the planktonic foraminifera $\delta^{18}O$ record and the new SST, since some of the heaviest isotope values (e.g. MIS 6 or MIS 14) (OK, it is true that could be due to either colder temperatures or very high salinity, or a combination of both factors) do not correspond with very low temperatures. I would also find interesting the discussion of IP25 record and the SST, to see if there are potential/expected biases due to the presence of sea ice which could have an effect on the diatom assemblages.

On reflection, we also conclude that the differences in SST are small and not significantly different across most of the record. We have changed the text accordingly and added some new biomarker-based estimates of seasonal sea ice conditions, which confirm the low amplitude environmental change at the site. It is also true (as 2 reviewers hint at) that G/IG changes are not as large at high latitudes as mid-low latitude settings.

Is panel Fig. 1c necessary since it is shown again in Figure 2 (plus three additional proxy curves?).

We consider it helpful to have the SST data from the full suite of short cores here so that they can be viewed alongside the maps; figure 2 permits a comparison between an array of different temperature records and the nuances of alkenone-based SST reconstructions where %C37:4 is elevated.

In figure 4a, the use of green and red in the proxy curves will make things difficult to color-blind people. Please use other colors.

Changed throughout

Same for figure 5b with brown, red and green.

Changed throughout

Figure 6. I am not sure if it is a formal convention in isotope geochemistry in paleoceanography, but I would ask the authors to invert the axis of the LR04.

Reversed as requested

References

- Dunkley Jones, Tom, et al. "OPTiMAL: A new machine learning approach for GDGT-based palaeothermometry." *Climate of the Past* 16.6 (2020): 2599-2617.
- Harada, Naomi, et al. "Characteristics of alkenones synthesized by a bloom of *Emiliana huxleyi* in the Bering Sea." *Geochimica et Cosmochimica Acta* 67.8 (2003): 1507-1519.
- Ishii et al. "New isoprenoid GDGT index as a water mass and temperature proxy in the Southern Ocean." *EGUsphere* 2025 (2025): 1-24.
- Kaiser, Jérôme, et al. "Changes in long chain alkenone distributions and Isochrysidales groups along the Baltic Sea salinity gradient." *Organic Geochemistry* 127 (2019): 92-103.
- Novak, Joseph, et al. "U38MEK' Expands the linear dynamic range of the alkenone sea surface temperature proxy." *Geochimica et Cosmochimica Acta* 328 (2022): 207-220.
- Park et al. "Seasonality of archaeal lipid flux and GDGT-based thermometry in sinking particles of high-latitude oceans: Fram Strait (79 N) and Antarctic Polar Front (50 S)." *Biogeosciences* 16.11 (2019): 2247-2268.
- Tierney, J. E., & Tingley, M. P. (2018). BAYSPLINE: A new calibration for the alkenone paleothermometer. *Paleoceanography and Paleoclimatology*, 33, 281–301.
<https://doi.org/10.1002/2017PA003201>
- Tierney, Jessica E., and Martin P. Tingley. "A Bayesian, spatially-varying calibration model for the TEX86 proxy." *Geochimica et Cosmochimica Acta* 127 (2014): 83-106.

Reviewer #3 (Remarks to the Author):

This manuscript presents downcore data for a new biomarker-based temperature proxy for use in cold water settings. It's great to see new proxies being developed in these challenging settings, and EZ25 has the potential to be a really important component of polar paleoclimate work. It's encouraging to see that the temperatures reconstructed are for the most part realistic and align with other data. However, I have several concerns about the paper as it's currently presented.

Firstly, as detailed more below, there are numerous errors, especially throughout the figures where some careful revising and checking is required.

As per the other two reviewers, we apologise for these errors and have made the necessary changes.

Secondly, I understand that an initial paper has just been published presenting this proxy in Organic Geochemistry, but this manuscript currently reads as if it is just an extension of that, rather than a new paper published in a different journal. I found myself needing to repeatedly go back to the Organic Geochemistry paper for clarification or to see if questions I had were answered in there. Some degree of that is always expected when reviewing a manuscript, but in this case I think more work is required to better ensure this paper stands alone. In particular, the results and site specific interpretations of the down core records are described, but the discussion as to what all these results mean for the proxy is limited. What do these results suggest for the impact of seasonality on the proxy? Or other factors that can often confound other proxies, like changes in species present through time, salinity, nutrients, surface or subsurface signal etc? The outlook section is very brief, what sort of next steps would help to improve the proxy or determine aspects like the upper temperature limit, or how far back in time the proxy could go to, or if species other Rhizosolenia could be producing the constituent HBIs?

This sounds like a proxy with a lot of potential, and it's important to see a more thorough discussion and presentation of ideas for refinement following the description of results.

This manuscript was originally written for other journals within the Nature portfolio, which have a reduced word limit compared to Communications Earth and Environment. As such, it was only possible to present the SST data with little scope for discussion of possible limitations or indeed of the other questions raised by this reviewer. At this stage it is not possible to answer all of these, but we recognise that some re-iteration of the factors first raised in Belt et al. (Organic Geochemistry, 2025) would be beneficial for the current paper and would provide a useful scene-setter for future studies, some of which are currently underway in our laboratories and will be reported on in due course. That said, we prefer to keep this additional text relatively short so as not to detract from the impact of the first paleo SST datasets that are presented herein, which remains our primary focus.

Some line/figure comments below.

Line 43: Sentence starting 'For', suggest for readability this sentence is split into two as it's quite hard to follow.

This has been updated as per Reviewer 2.

While I definitely agree there are challenges with applying a GDGT-based paleothermometer at lower temperatures, the description here an oversimplification that doesn't acknowledge there's been a lot of work on better calibrating polar GDGTs. For instance spatially varying or regional specific calibrations, or OH-GDGT-based proxies (i.e. Fietz et al., 2016 <https://doi.org/10.1016/j.orggeochem.2016.10.003>, Varma et al., 2024,

<https://doi.org/10.1016/j.gca.2023.12.019>). I think your point can still hold by acknowledging that a lot of work has been done on polar temperatures proxies, but there are still challenges to the current methods we try to use at high latitudes, and that any new proxy that can better reconstruct polar temperatures is very welcome.

It was not our intention to oversimplify or understate the value of other SST proxies, but again, the reduced (original) word limit prevented more detailed background material. We have modified the text accordingly

Line 56: Correct format of UK37.

Changed, as requested.

Line 147: Remove 'in' from before JM99-1200.

Changed, as requested.

Lines 197-201: See comments about Figure 7.

Updated as above

Line 365: What supplementary information is this referring to? Supplementary Table 1 doesn't appear in the review document? Will the new data that is referred to in this paper be included in a supplementary table too?

Yes, all data are available, as per the journal guidelines. A data availability statement has been added

Methods: This jumps around a bit and it would help if it was more structured i.e. shift to the top what core data has been previously reported, and what is new. With the exception of the age models for the box cores developed here, where have the age models been reported for the previously studied cores, and cores that other new data has been analysed from?

We have re-ordered the Methods section, as requested

Figures all need to be carefully checked.

Agreed!

Figure 1: c) and d) are labelled around the wrong way. Is the black dot on c) the modern temperature for core 12?

Figure 2: a) and b) are described the wrong way around in the caption.

These have been switched and the legend for (c) has been updated to explain)

The panel looks to be the same as panel b) from Figure 1. Suggest that these figures are reworked so the same panel isn't shown twice, as well as ensuring the captions are correct.

Please see response to Reviewer 2 for our rationale to include some of the data plots twice.

Figure 3 caption describes core 13 as in panel a) but the figure has it numbered on panel b).

Cores 13 and 16 have been switched in the legend.

Figure 4 describes an orange line in the caption but I believe this in blue in the figure.

Indeed, legend changed, as indicated.

Figure 5 has swapped the order of a)-e) from bottom to top, compared to the other figures where a) is at the top.

We have re-ordered the figure so that it is a) (top) to e) (bottom).

Panel b) describes an orange line instead of blue.

Changed throughout

d) looks to have a dark blue line rather than purple?

Colours have been updated throughout to increase clarity

Figures 6 and 7: Convention is that the LR04 stack is shown with higher values on the bottom i.e. with the y axis reversed. In particular on Figure 7, having this axis reversed makes it quite hard to read the figure, especially as the $\delta^{18}O$ from the core itself is shown the opposite.

While the reconstructed temperature appear appropriate, there does not seem to be a particularly clear relationship between warmer temperatures during interglacials and cooler in glacials. The wording of lines 197 to 201 in the text glosses over this and instead only highlights cool temperatures in MIS 16 and warm 'post LGM', when there's also cool data at the height of MIS 11, and the warmest sample actually looks to be in the LGM. I'd prefer this was more clearly discussed, and instead used as a good opportunity to discuss some potential reasons for this variability and lack of clear glacial/interglacial pattern (i.e. could this be to do with the site-specific environment or is there some interglacial/glacial change in something like the diatom HBI producer that could impact how the proxy works).

As per reviewer 2, we have revised the text to cover this and added new data (SpSIC) which supports the low amplitude SST changes. We have changed the isotope data plots as requested

Responses to Reviewers (**in bold**)

Once again we thank the reviewers for their feedback

Reviewer #1 (Remarks to the Author):

2nd Review of Belt et al. for Nature Communications Earth and Environment
Joseph B. Novak

Summary

The revised version of the manuscript presented by Belt et al. addresses my major concerns. I look forward to seeing this work published.

Minor Comments

L6: Typo here.

It's not clear to us where the typo is, After discussion with the Editor, no change has been made

L305: The title of reference 12 is incorrect.

Corrected as requested

Reviewer #2 (Remarks to the Author):

This is the second time I have seen this manuscript. Overall, I think that it is much improved but could still use some editing. The authors present an interesting interpretation of their data in the context of the EZ25 data. I am satisfied with how the authors have dealt with reviewer comments. I only have a couple of comments:

I will think that the authors should include standard error in the figures, even if this error does not include the error propagation (analytic errors). It is meaningful to see whether some of the changes in the EZ25 data are meaningful (larger than the calibration error) or not. This is important for example in ODP Site 910, where the authors acknowledge that the SST changes are too small to be taken into account.

The standard error has been added to all data figures. We have also added the SE for other proxies

L. 74: "makes selection of the most appropriate, problematic". I do not really agree that selection is problematic, and the data can be reconstructed using different calibrations...

The problem, as we see it, is that calibration selection is not always justified (apart from the one that fits best). More importantly, different calibrations can yield quite different outcomes (see the examples of the different UK37 SSTs shown herein for core MSM05-712-1; Figure 2a). As such, making meaningful

comparisons between SST data from different studies/calibrations can be challenging. We have amended the text to better convey this point. The inclusion of this point is primarily to provide some background context to what we present in terms of a single calibration for both polar regions. We revisit this in the Outlook section.

Reviewer #3 (Remarks to the Author):

Thank you to the authors for revising the manuscript and correcting the figure errors. My comments from the initial review have been mostly addressed and the manuscript has been improved. Including topic sentences in the results and discussion has improved the readability of the section. The outlook section has also been expanded and now includes a more thorough discussion of potential confounding factors for the proxy as well as future research directions. A couple of small comments.

Lines 44-49: Suggest this sentence is split into two for readability.

Revised as requested

Lines 71-79: Similar to a comment by another reviewer, I think adding a sentence in here about the standard error of the proxy in the water column (and why you have not applied it to the sediment data) is important context for interpreting the results. It would be good to still see this section expanded a bit more to better introduce the proxy, i.e. adding a more specific upfront comment on the potential diatom source for this proxy, rather than waiting to discuss this in the outlook section.

The SE for the proxy (and other proxies) has been added to each figure and a description added to each figure legend.

The likely diatom source has been added and the end of the Introduction slightly modified to make it clearer that the main aim here was to test the EZ25 index as a palaeo SST proxy by measuring it in sediments. We believe this now makes the link clearer from the previous work (water column) to the current study. The SE issue is also mentioned further in the outlook section.

Line 280: I recommend a conclusion statement is added to the end of the section as a final summary, as the newly added text finishes rather abruptly. I appreciate the addition of this outlook information, it sets the scene for future application of the proxy more clearly.

A final summary statement has been added, which links back to the article title

In going through final checks, we noticed that two SSTs in Figure 1 (cores 3 and 4) were originally reported as 9.8 and 7.2 °C, respectively. However, these are the mean values for the core, not the timeframe reported (since 1850). We have thus updated the values to 10.6 and 7.1 °C, respectively. The change does not alter the interpretation or significance.